# Optimal Underwater Acoustic Warfare Strategy Based on a Three-Layer GA-BP Neural Network

**DOI:** 10.3390/s22249701

**Published:** 2022-12-11

**Authors:** Zirui Wang, Jing Wu, Haitao Wang, Huiyuan Wang, Yukun Hao

**Affiliations:** 1School of Mechanical Engineering, Xi’an Jiaotong University, Xi’an 710000, China; 2Shanghai Electronic Ship Research Institute, China Shipbuilding Industry Corporation, Shanghai 201100, China

**Keywords:** GA-BP neural network, underwater acoustic warfare strategy, feedback system, defense platform, incoming torpedoes

## Abstract

A defense platform is usually based on two methods to make underwater acoustic warfare strategy decisions. One is through Monte-Carlo method online simulation, which is slow. The other is by typical empirical (database) and typical back-propagation (BP) neural network algorithms based on genetic algorithm (GA) optimization, which is less accurate and less robust. Therefore, this paper proposes a method to build an optimal underwater acoustic warfare feedback system using a three-layer GA-BP neural network and dropout processing of the neural network to prevent overfitting, so that the three-layer GA-BP neural network has adequate memory capability while still having suitable generalization capability. This method improves the accuracy and stability of the defense platform in making underwater acoustic warfare strategy decisions, thus increasing the survival probability of the defense platform in the face of incoming torpedoes. This paper uses the optimal underwater acoustic warfare strategies corresponding to incoming torpedoes with different postures as the sample set. Additionally, it uses a three-layer GA-BP neural network with an overfitting treatment for training. The prediction results have less error than the typical single-layer GA-BP neural network, and the survival probability of the defense platform improves by 6.15%. This defense platform underwater acoustic warfare strategy prediction method addresses the impact on the survival probability of the defense platform due to the decision speed and accuracy.

## 1. Introduction

In modern underwater acoustic warfare, the ability of the defense platform to accurately and quickly select the strategy with optimal effectiveness directly affects the survivability of the defense platform [1,2]. Among underwater acoustic warfare, noise jammers and acoustic decoys are the most widely used [3,4]. The noise jammer reduces the signal-to-noise ratio of active acoustic self-guided torpedoes by increasing their noise. Thus, the jammer has a decoy performance when it is far from the passive acoustic self-guided torpedo and causes saturation blockage of its receiver at close range. Acoustic decoys can be launched to a certain distance from the defense platform to lure the torpedo. The torpedo will be led to a different course from the defense platform. Figure 1 shows the defense platform’s underwater acoustic warfare posture. Although underwater acoustic warfare currently has acceptable performance, in many cases, a poor warfare strategy fails to serve the purpose of defense and leads the torpedo to the defense platform [5]. Hence, the strategic decision is crucial to the underwater acoustic warfare of the defense platform.

Many scholars have recently researched the underwater acoustic warfare strategies of defense platforms in recent years. Zhan K et al. [6] simulated the survival probability of defense platforms using different acoustic decoys and jammers as warfare strategies against different postures of incoming torpedoes by the Monte-Carlo method. They conducted genetic algorithm iteration to obtain the optimal underwater acoustic warfare strategy of defense platforms against each posture of the incoming torpedo. Chen Y C et al. [7] used a linear programming approach to find the optimal acoustic decoy, jammer launching course, and defense platform avoidance course for defense platforms facing different postures of the incoming torpedo. Zhang Y et al. [8] analyzed the characteristics of close-range underwater acoustic warfare and studied the optimal warfare strategy, including acoustic decoy, jammer launching course, and defense platforms avoidance course. However, these methods require a long decision time and cannot react the first time, affecting the defense platform’s survival probability in actual combat.

Currently, prediction neighborhoods are increasingly using machine learning and deep learning. Tie Zhang et al. [9] proposed a method based on the characteristics of intestinal sound signals to predict intestinal bowel movements using BP classification neural networks to achieve a noninvasive collection of physiological signals. Jumin Zhao et al. [10] addressed the critical problem of large sensor data volume and the consequent need to increase network throughput. Hence, they proposed a method for backward propagation. Yuanzhou Zheng et al. [11] proposed a path prediction model (GA-ACO-BP) combining a genetic algorithm, ant colony algorithm, and a BP neural network to address the critical problems of prediction accuracy and computational efficiency of the future position ship. Dong Lv et al. [12] discussed the lack of real-time service (RTS) provided by the International GNSS Service (IGS) product or network failure, which make RT-PPP unusable. Thus, they proposed an improved clock offset prediction model based on heterogeneously integrated learning and dynamic multigroup particle swarm optimizer (HPSO-BP) optimization.

Therefore, in this work, according to different incoming wire-guided and acoustic homing torpedo postures, a simulation was conducted to obtain the avoidance course, the rocket-assisted jammer, and the acoustic homing decoy launching course with the highest survival probability for the defense platform. Furthermore, we designed a three-layer GA-BP neural network to establish an optimal underwater acoustic warfare strategy feedback model to improve the decision speed and accuracy of the defense platform in formulating a warfare strategy.

## 2. Materials and Methods

Usually, after a defense platform detects an incoming torpedo, it develops a warfare plan. It launches a noise jammer to convert the enemy platform to the acoustic homing mode by disconnecting it from the torpedo. When the jammer enters the water and starts working, the defense platform evades and launches an acoustic decoy to make earlier acoustic contact with the torpedo.

### 2.1. Defense Platform Maneuver Evasion Strategy

After the launch of the jammer, the defense platform maintains a uniform and straight course. After the jammer enters the water and starts to work, the ship must choose the optimal course and speed to maneuver to avoid the incoming torpedo [13]. Figure 2 shows the evasion strategy of the defense platform, setting the torpedo to cut off the guide wire at *T*_1_ and turn to acoustic homing. The enemy angle is *α*, the distance from the ship is *L*_1_, and the course angle is *β*. Moreover, the defense platform anticipates that the torpedo will turn on homing when it arrives at point *T*_2_. The distance from this point to the expected target position of the ship is *L*_2_. When the ship takes evasion, its optimal route is along a straight line away from the direction between points O and *T*_2_.

By the position *T*_1_*(X*_1_*, Y*_1_*)* at point *T*_1_, and projecting the position *T*_2_*(X*_2_*, Y*_2_*)* at point *T*_2_, the optimal maneuvering avoidance course *C_θ_* for the ship is:(1)$$C_{\theta }=\{\begin{array}{l} \arctan(\frac{Y_{2}}{X_{2}}),\;T_{2}\;\mathrm{in}\;\mathrm{quadrant}\;2\;\mathrm{or}\;3\\ \arctan(\frac{Y_{2}}{X_{2}})+180^{\circ },\;T_{2}\;\mathrm{in}\;\mathrm{quadrant}\;1\;\mathrm{or}\;4 \end{array}$$

### 2.2. Jammer Deployment Strategy

The defense platform launches the jammer at a maximum distance, and its optimal launch position is on the azimuth line with the enemy launch platform [14]. Figure 3 shows the jammer deployment strategy, with the location of the defense platform at the time of torpedo alarm set as the coordinate origin, and the defense platform sails at a constant speed with course *C_W_* and speed *V_W_*. The torpedo alarm is at point *O*, the defense platform finds the enemy platform at point *M*, and the course angle of the enemy platform is *Q_M_*. After the system decision reaction time *t_sys_*, it reaches point *W_1_*. It launches the jammer at that point, and the positions of the defense platform and the enemy platform when the jammer enters the water are points *W_e_* and *M_e_*, respectively. Then, the optimal landing point of the jammer is point *J*, and *d_f_* is the maximum range of the jammer.

Firstly, the jammer entry point is on a circle with *W*_1_ as the center and *d_f_* as the radius. Hence, obtaining the limiting equation for the location of the jammer entry point is as follows:(2)$${\left(x_{J}-V_{W}\cdot t_{sys}\right)}^{2}+y_{J}{}^{2}=d_{f}{}^{2}$$

Next, the jammer entry point is between points *W_e_* and *M_e_*. So, we can obtain the same from the limiting equation of the jammer entry point position:(3)$$y_{J}=(x_{J}-V_{W}\cdot (t_{sys}+\frac{d_{f}}{V_{f}}))\cdot \frac{\left(y_{M}+\sin(Q_{M})\cdot V_{M}\cdot (t_{sys}+\frac{d_{f}}{V_{f}})\right)}{\left(x_{M}+\cos(Q_{M})\cdot V_{M}\cdot (t_{sys}+\frac{d_{f}}{V_{f}})-V_{W}\cdot (t_{sys}+\frac{d_{f}}{V_{f}})\right)}$$

Finally, we can obtain the jammer’s entry point *J* by concatenating Equations (2) and (3), thus obtaining the jammer’s emission direction *C_J_*.

### 2.3. Acoustic Decoy Deployment Strategy

The direction of the acoustic decoy launch should ensure that the torpedo finds the acoustic decoy first and deceives the torpedo to deviate from the tracking course. Figure 4 shows the acoustic decoy deployment strategy. The strategy is to set the location of the defense platform at the time of the torpedo alarm as the origin of the coordinates, the defense platform’s initial course as *C_W_*, and the enemy’s angle as *α*. The torpedo is located at point *T*. The acoustic decoy is launched, and the direction of *C_θ_* is turned to maneuver to evade the torpedo.

Usually, defense platforms launch an acoustic decoy in three ways. The first one launches on the initial course *C_W_* of the defense platform, the second one launches in the direction *OT* of the torpedo, and the third one launches in the direction *V_d_* between the direction of the torpedo and the initial course of the defense platform [15,16]. When the acoustic decoy is launched simultaneously with the jammer, care is needed to avoid the acoustic decoy being interfered with by the jammer and failing to make acoustic contact with the torpedo before the defense platform.

### 2.4. Back Propagation Neural Network Theory

A back propagation (BP) neural network is a multilayer artificial neural network that back-propagates the feedforward error, which adjusts the network to minimize the error sum of squares by multiple error feedback [17,18], so that the network can map the input and output of known data. Therefore, BP neural networks can fit arbitrary nonlinear mappings and have strong generalization and adaptation capabilities [19]. The typical BP neural network consists of three parts: the input layer, the hidden layer, and the output layer. The topology of the typical BP neural network is shown in Figure 5.

Among them, *x*_1_*, x*_2_*, ··· x_m_* are the input values, *y*_1_*, y*_2_*, ··· y_n_* are the predicted values, *ω_ij_* is the connection weight between the input layer and the hidden layer, and *ω_jl_* is the connection weight between the hidden layer and the output layer. *m*, *k*, and *n* are the nodes in the input, hidden, and output layers. *a* and *b* are the threshold values in the hidden layer and the output layer, respectively, and *f* is the incentive function of the hidden layer.

Firstly, the hidden layer output is:(4)$$H_{j}=f\left(\sum_{i=1}^{m}\omega_{ij}x_{i}-a_{j}\right),j=1,2,\cdots ,k$$

Then, the output layer output is:(5)$$O_{l}=\sum_{j=1}^{k}H_{j}\omega_{jl}-b_{l},l=1,2,\cdots ,n$$

Next, the error between the network and the expected output is:(6)$$e_{l}=y_{l}-O_{l},l=1,2,\cdots ,n$$

Finally, the weight and the threshold are updated to:(7)$$\{\begin{array}{l} \omega_{ij}=\omega_{ij}+\eta H_{j}(1-H_{j})x_{i}\cdot \sum_{l=1}^{n}\omega_{jl}e_{l}\\ \omega_{jl}=\omega_{jl}+\eta H_{j}e_{l} \end{array}$$
(8)$$\{\begin{array}{l} a_{j}=a_{j}+\eta H_{j}(1-H_{j})\cdot \sum_{l=1}^{n}\omega_{jl}e_{l}\\ b_{l}=b_{l}+e_{l} \end{array}\infty$$

### 2.5. Genetic Algorithm to Optimize BP Neural Network Method

The genetic algorithm (GA)-optimized BP neural network has the advantages of fast convergence and the ability of global optimization seeking compared with the typical BP neural network. Moreover, we can obtain the initial weights and thresholds of the BP neural network through GA, which makes the prediction results of the neural network closer to the actual results [20,21,22]. Figure 6 shows the specific process.

Its specific steps are as follows.

Input normalized training data.Determine the BP neural network topology.Encode using the real number encoding method and initial each chromosome in the population, which includes the connection weights and thresholds of the BP neural network, whose chromosome length *L* can be derived by the following equation:
(9)L=lm+ln+m+n
Among them, *l*, *m*, and *n* are the nodes in the input layer, the hidden layer, and the output layer.Substitute the initialized network weights and thresholds into the BP neural network. We can obtain the fitness value of each chromosome by the absolute value of the difference between the predicted and the actual results.Genetic evolution is achieved by the roulette wheel method for selection and the real number crossover method.We judge whether the genetic evolution reaches the set maximum times, and if the genetic evolution reaches the maximum evolution, the GA stops running and obtains the last generation population. Otherwise, it returns to step 5.We decode the chromosome with the optimal fitness value from the last generation population in GA and substitute it into the BP neural network to obtain the initialized connection weights and thresholds to start the training. The BP neural network completes training when the network reaches the set accuracy or the maximum number of iterations.The output obtains the prediction results of the network model.

## 3. Results and Discussion

### 3.1. Optimal Warfare Strategy Planning for Defense Platforms under Monte-Carlo

The collection of training data is crucial to the training effect of the neural network; thus, sample data must be complete and comprehensive. At the same time, the sample data should avoid deterioration of the network generalization ability due to excessive complexity.

Since the defense platform develops the avoidance course, jammer, and acoustic decoy launch direction according to the distance, angle, and speed of the incoming torpedo, the effect is closely related to the incoming torpedo posture. The homing acoustic decoys are deployed in a certain range (between the direction of the incoming torpedo and the initial course). We must combine its deployment according to the incoming torpedo posture and how we deploy the jammer. We simulate the warfare strategy under different incoming torpedo postures by the Monte-Carlo method. We simulate each posture separately 100 times and obtain the optimal underwater acoustic warfare strategy based on the effectiveness assessment [23,24,25].

Figure 7 shows the optimal underwater acoustic warfare strategy simulation method, simulated by three modules: defender, attacker, and effectiveness evaluation.

Figure 8 shows the simulation flow of the defense side module. Firstly, the heading and speed of the defense platform need to be set, and the information on the location where the defense platform detects the torpedo must be gathered. Secondly, after the defense platform detects the incoming torpedo, it launches the jammer after the decision time of the warfare strategy. It enters the new navigation according to the initial course and speed to avoid the torpedo discovering the evasive intention of the defense platform. Finally, when the jammer enters the water, the defense platform evades by adjusting the course and launches an acoustic decoy to lure the torpedo until it deviates from the tracking course. From this, it is clear that the decision time of the defense platform to make the counter strategy affects the timing of the jammer launch and thus the timing of the defense platform evasion and acoustic decoy launch.

Moreover, the course of the jammer launch affects its effectiveness in masking the enemy platform from detecting the defense platform. The avoidance course of the defense platform affects the probability of being detected by the torpedo and the range of the torpedo tracking it. The acoustic decoy launch course affects the ability of the acoustic decoy to make contact with the torpedo before the defense platform and also affects the ability of the torpedo to detect the acoustic decoy without being obscured by the jammer. Therefore, the effectiveness of the defense platform in developing warfare strategies and the speed of decision making are crucial to its survival probability.

Figure 9 shows the simulation flow of the attacker module. First, it is necessary to set the initial course and speed of the torpedo and track the defense platform according to the wire-guided mode. Secondly, it is necessary to judge whether its interference sector makes the attacking platform lose the target after the jammer enters the water. If the target is lost, the torpedo cuts off the wire guide to acoustic homing. If no target is lost, the wire-guided way tracks the target until it reaches the acoustic homing opening distance. Then, the torpedo detects the target. If the target is detected, the target is tracked to the recognition distance when the torpedo recognizes the target. If the torpedo identifies it as a true target, then it attacks; if the torpedo recognizes it as a false target, it conducts circular research to find the next target until it runs out of range.

Figure 10 shows the simulation process of the effectiveness evaluation module. First, the simulation simulates a single underwater acoustic combat process. The second step is to judge the number of simulations on whether they achieve the set value. If the number of simulations achieves the set value, the probability of a torpedo hit and the defense platform’s survival probability are calculated. If the number of simulations does not reach the set value, the number of simulations is increased, the simulation parameters are initialized, and random errors are generated.

When setting the target parameters, we set the active acoustic homing torpedo to use the fixed advance angle guidance method and set the fixed advance angle to 5°. Since torpedoes usually attack from the side and rear of the target, we set the initial enemy side angle to 135°. We set the submarine and mobile acoustic decoy line array length as 60 m, and both mobile and suspended acoustic decoy echoes adopt the highlight simulation method. Submarine echoes adopt the plate element simulation method.

When setting the signal parameters, we set the operating frequency of the torpedo launch signal to 30 kHz and the bandwidth to 0.8 kHz. The signal period is 1 s, the time width is 62.5 ms, the sound source level is 220 dB, the noise level is 60 dB, the received directivity index is 17 dB, and the detection index is 12 dB.

In setting the ocean environment, we applied the sound velocity gradient derived by Dushaw from summer temperature, pressure, and salinity data in the North Atlantic Ocean off the coast of the U.K. and Ireland. We set the ocean’s depth to 600 m. The source and receiver points’ depth was 200 m, the source signal’s emission angle range was −15°–15°, and the source emitted 20 signals at equal intervals. The receiving point receives the signal in its depth range of −10–10 m, and the receiving angle range is also −15°–15° [26].

By simulation, we can see that when the target is within 500 m of the active acoustic homing torpedo, the torpedo can recognize the suspended acoustic decoy as a point target. At the same time, the mobile acoustic decoy does not have a highlighted background, and the torpedo can recognize it as a fake target. Figure 11 shows the echoes of the submarine, mobile acoustic decoy, and suspended acoustic decoy at a 500 m distance. Therefore, we can set the active acoustic homing torpedo to recognize the fake target at 500 m.

We set the initial course of the defense platform as 0°, the initial speed as 18 kn, the speed at evasion as 30 kn, the rotational angle speed as 2°/s, and the speed as 15 kn. The speed of the homing-type acoustic decoy is 15 kn. The rocket-assisted jammer drop distance is 2.3 km, and the launch to water time is 20 s. Moreover, the remaining range of the torpedo is 15,000 m, the rotational angle speed is 10°/s, and the dummy target identification distance is 500 m. The speed of the enemy platform is 8 kn; the distance is 10,000 m. The torpedo alarm sonar has a certain error, so the estimated torpedo speed error is 2 kn, the torpedo angle error is 10°, the torpedo distance error is 300 m, and the system decision reaction time is negligible.

Although we can obtain the optimal underwater acoustic warfare strategy by the Monte-Carlo method, its decision time is long and requires strict hardware and software performance. To speed up the decision time, we used a GA-BP neural network to predict the optimal underwater acoustic warfare strategy so that the defense platform can almost ignore the decision time. By this method, the defense platform can make defense measures the first time to minimize the impact of decision speed on defense effectiveness.

In order to simulate the incoming torpedo under different postures, the simulated torpedo speed is 40–50 kn with a step of 1 kn. The ramp angle is 0°–180° with a step of 5°. The distance is 5000–10,000 m with a step of 50 m, and we can obtain 37,000 data samples. Table 1 shows the sample set of neural network training and test data.

Among them, *X* is the incoming torpedo distance data, *Y* is the incoming torpedo angle data, and *Z* is the incoming torpedo speed. Moreover, *B* is the corresponding optimal strategy, which includes the avoidance course, jammer launching course, and acoustic decoy launching course.

We disrupt the data samples and take 30,000 data samples as training data. Correspondingly, we take the remaining 7000 data samples as test data. Figure 12 shows the optimal warfare strategy simulation.

### 3.2. Normalization of Values

Firstly, since there are differences in the order of magnitude of the input data of the training samples, the network weights will generate large errors when the differences are large, affecting the model’s accuracy and convergence speed. Therefore, the input data are normalized to reduce the network weight errors caused by the order-of-magnitude differences and improve the accuracy of the prediction model. In this paper, we restrict the input data to the interval [0,1], and the normalization process is as follows:(10)$$X=\frac{\left(x-x_{\min}\right)}{\left(x_{\max}-x_{\min}\right)}\cdot (upper-lower)+lower$$

In the formula, *x*_max_ is the input data’s maximum value, *x*_min_ is the input data’s minimum value, *upper* is 1, and *lower* is 0.

Next, we evaluate the network model by comparing the error between predicted and actual values. The output results need to be normalized, and the normalization process is as follows:(11)$$Y=\frac{\left(y-lower\right)}{\left(upper-lower\right)}\cdot (y_{\max}-y_{\min})+y_{\min}$$

In the formula, *y*_max_ is the maximum value of the output data; *y*_min_ is the minimum value of the output data.

### 3.3. Setting of Network Training Parameters

The GA-BP neural network training parameters affect the error and convergence speed of network training [27], and the activation function functions of the implicit layer and output layer in the simulation are chosen as “Relu-Leaky” and “purelin”, respectively. The leaky function is convenient for derivative calculation, can converge faster, has one-sided suppression and a wider boundary, and can obtain the gradient when the network input is negative [28]. We chose “trainlm” as the backpropagation training function; the learning rate was 0.01, the target number of iterations was 1000, and the target error was 10^−6^. The genetic algorithm population size was 50, the crossover probability was 0.3, the variance probability was 0.15, and the number of genetic generations was 100.

### 3.4. Confirmation of Optimized GA-BP Neural Network Model

We evaluated the performance (fitness) of the GA-BP neural network model according to the individual test errors, where *MAE* is the mean absolute error, *MAPE* is the mean absolute percentage error, and *RMSE* is the root mean square error [29]. The formula for calculating each error is as follows:(12)$$V_{MAE}=\sum_{i=1}^{N}(\sum_{j=1}^{M}\left|y_{ij}-{\tilde{y}}_{ij}\right|\cdot \frac{1}{M})\cdot \frac{1}{N}$$
(13)$$V_{MAPE}=\sum_{i=1}^{N}(\sum_{j=1}^{M}\left|\frac{y_{ij}-{\tilde{y}}_{ij}}{y_{ij}}\right|\cdot \frac{1}{M})\cdot \frac{1}{N}$$
(14)$$V_{RMSE}=\sqrt{\sum_{i=1}^{N}{\left(\sum_{j=1}^{M}\left|y_{ij}-{\tilde{y}}_{ij}\right|\cdot \frac{1}{M}\right)}^{2}\cdot \frac{1}{N}}$$

In the formula, *y_i_* is the actual value, *ỹ_i_* is the predicted value, *M* is the output quantity of the neural network, and *N* is the number of test sample data.

Choosing the appropriate number of implied layers and nodes is especially critical to the network’s performance. The increase in the number of implied layers and nodes can improve the memory capacity of the network. However, when the number of implied layers and the number of nodes in each layer is too high, it can cause overfitting of the network. Additionally, we are managing the memory hardware according to the power of 2 since the engineering implementation is considered. We analyzed the network performance for the number of implicit layers 1–3 and the number of nodes in the implicit layers 8, 16, 32, and 64. Among them, the implicit layer of 1 and the number of nodes of 8 are the typical single-layer GA-BP neural network structure.

Figure 13 compares the test errors of the typical GA-BP network and the optimized GA-BP network. We can see that the MAE, MAPE, and RMSE errors of the three-layer GA-BP neural network with dropout treatment are only 0.147, 0.0061, and 0.2016, respectively. It is slightly lower than the errors of the three-layer GA-BP neural network without dropout treatment and much lower than the errors of the two-layer and one-layer GA-BP neural network MAE, MAPE, and RMSE errors. The network memory ability becomes more robust as the number of hidden layers and the number of nodes in the hidden layer of the network increase. However, too many hidden layers and nodes in the network’s hidden layer will cause overfitting, the network generalization ability will decrease, and the test error will increase instead. The dropout process solved the overfitting problem, temporarily removing the neural network training unit from the network with a certain probability. Based on the research described in this paper, the optimal underwater acoustic warfare strategy uses a neural network of three-layer GA-BP with a topology of 8-16-64. Figure 14 shows the topology of the GA-BP neural network in the optimal underwater acoustic warfare strategy feedback system derived in this paper.

The simulation simulates the test results of the defense platform warfare strategy at a target speed of 45 kn. Figure 15 and Figure 16 show the differences between the actual output and the test output of the typical GA-BP neural network and the optimized GA-BP neural network, respectively. Figure 15a–c and Figure 16a–c show the differences between the actual output and test output of the defense platform avoidance course, the jammer launch course, and the acoustic decoy launch course, respectively.

We can see that the prediction error peaks of the defense platform evasive course, jammer launch course, and acoustic decoy launch course are smaller after optimization. The optimized GA-BP neural network can more accurately predict the optimal underwater acoustic warfare strategy. Moreover, the error undulation distribution is reduced after optimization. The optimized GA-BP neural network can better suppress the excessive prediction error of the optimal underwater acoustic warfare strategy under the chord angle of individual incoming torpedoes. Therefore, the optimized GA-BP neural network improves the defense platform’s decision accuracy, expands the defense platform’s range to make accurate decisions, and ensures that the defense platform can maintain high prediction accuracy even when making the worst decisions. Overall, the optimized GA-BP neural network improves the accuracy and stability of the defense platform in making underwater acoustic warfare strategy decisions.

Figure 17 shows the optimal underwater acoustic warfare strategy feedback system. Table 2 shows the performance comparison between the optimized and typical GA-BP neural networks. The survival probability of the defense platform using the two GA-BP neural networks to establish the underwater acoustic warfare feedback system for the defense platform was simulated and analyzed separately. We can see that the optimized GA-BP neural network performs better than the typical GA-BP neural network. The survival probability of the defense platform using the optimized GA-BP neural network to establish the underwater acoustic warfare feedback system for the defense platform reached 94.34%, which improved the survival probability of the typical GA-BP neural network defense platform by 6.15%. Therefore, the optimal underwater acoustic warfare strategy feedback system for the defense platform based on the three-layer GA-BP neural network designed in this paper enables the defense platform to improve its defense capability against torpedo threats.

## 4. Conclusions

(1) In this paper, we address the problems of the slow speed of making underwater acoustic warfare strategy decisions by database or Monte-Carlo method online simulation for defense platforms and the poor accuracy and stability of making underwater acoustic warfare strategy decisions by database search or a typical single-layer GA-BP neural network. After consideration of the engineering hardware implementation, we propose a three-layer GA-BP neural network prediction model with dropout processing to establish an optimal underwater acoustic warfare strategy feedback system for incoming torpedoes of different postures.

(2) On the theoretical basis of defense platform avoidance, acoustic decoy, and jammer deployment, we used the Monte-Carlo method to simulate and obtain optimal underwater acoustic warfare strategies for incoming torpedoes in each posture. We also trained the neural network with this as the sample data. Finally, we compared the performance of the typical GA-BP neural network and the three-layer GA-BP neural network feedback system with dropout processing. The survival probability of the defense platform is improved by 6.15% when the defense platform uses the three-layer GA-BP neural network strategy feedback system with dropout processing. This method of predicting underwater acoustic warfare strategies for defense platforms minimizes the impact of decisions on the survival probability of defense platforms.

If the defense platform needs to improve the survival probability further, it needs to improve the acoustic decoy performance, torpedo alarm sonar performance, defense platform mobility, and defense platform stealthiness.

## Figures and Tables

**Figure 1 sensors-22-09701-f001:**
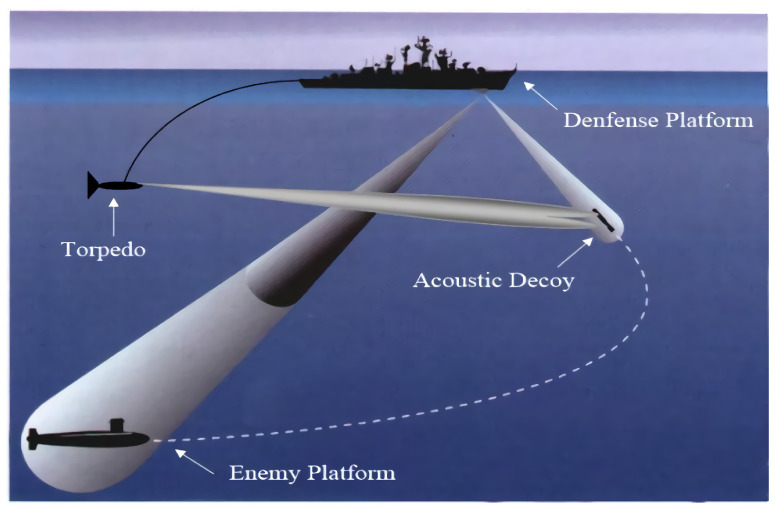
The defense platform’s underwater acoustic warfare posture.

**Figure 2 sensors-22-09701-f002:**
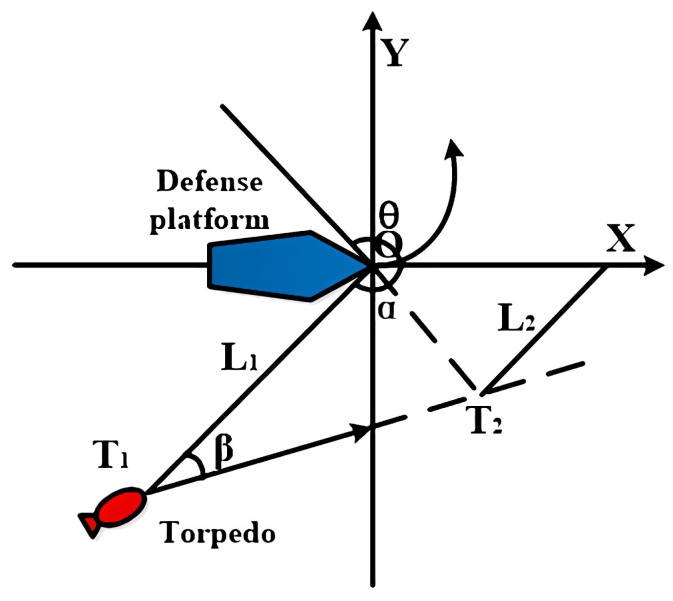
The defense platform avoidance scheme.

**Figure 3 sensors-22-09701-f003:**
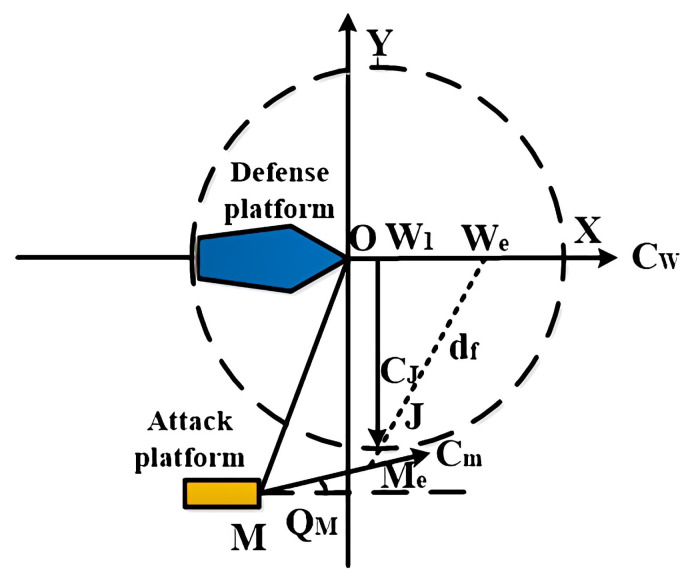
The jammer deployment scheme.

**Figure 4 sensors-22-09701-f004:**
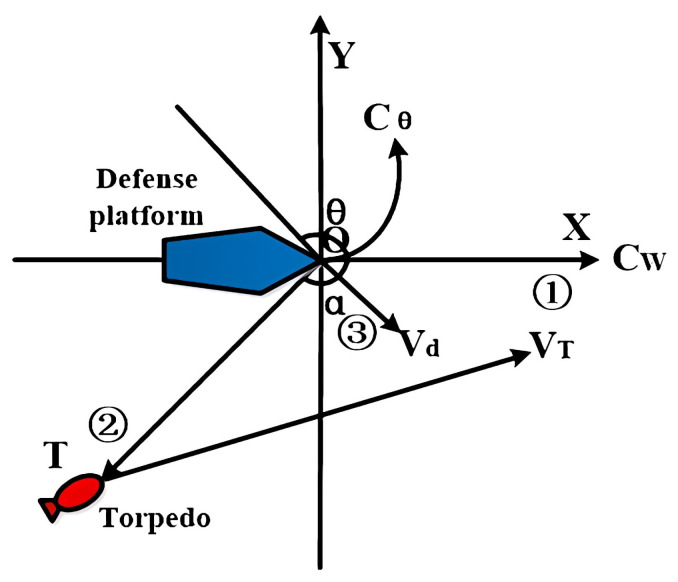
The acoustic decoy deployment scheme.

**Figure 5 sensors-22-09701-f005:**
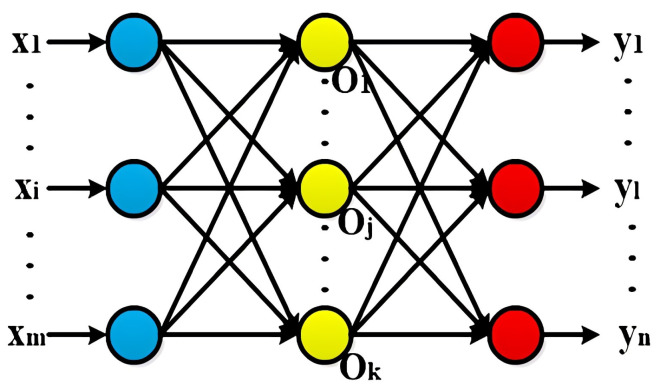
The topology of the BP network.

**Figure 6 sensors-22-09701-f006:**
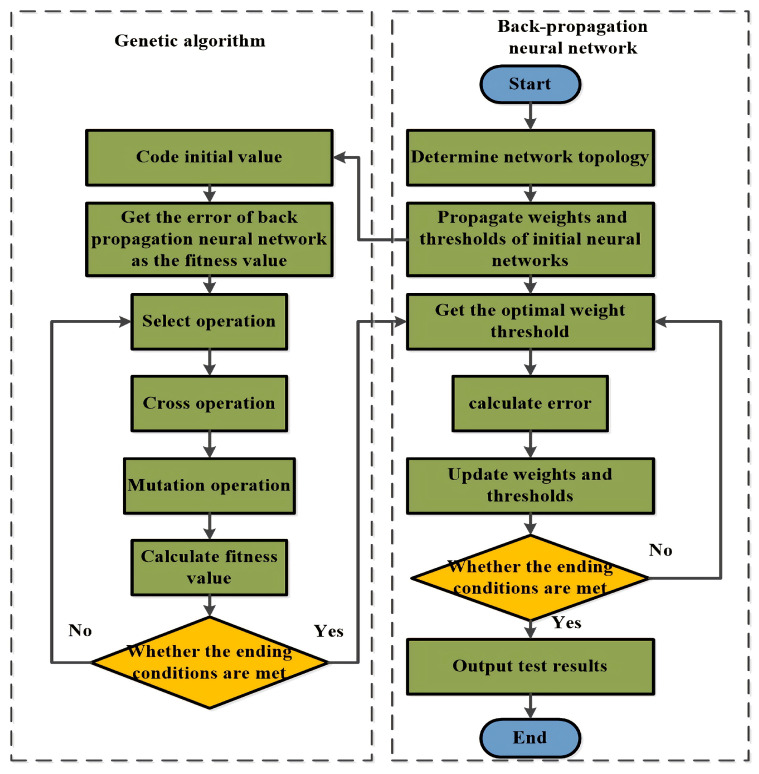
The GA-BP module process.

**Figure 7 sensors-22-09701-f007:**
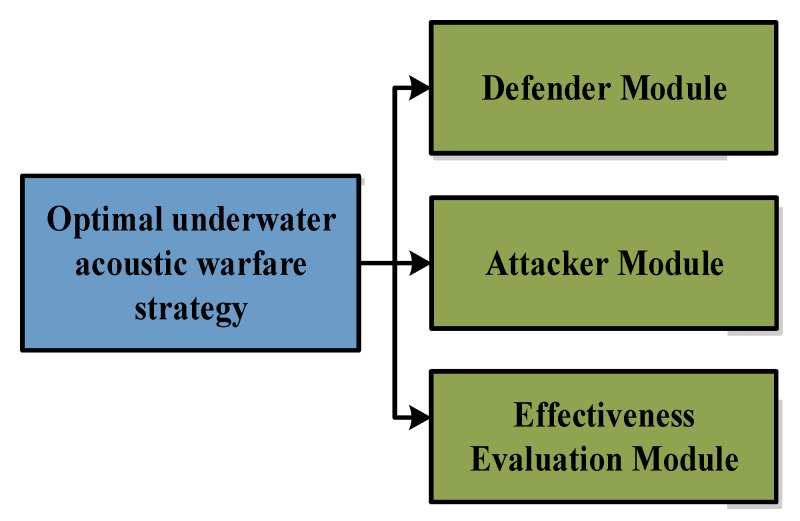
The optimal underwater acoustic warfare strategy development method.

**Figure 8 sensors-22-09701-f008:**
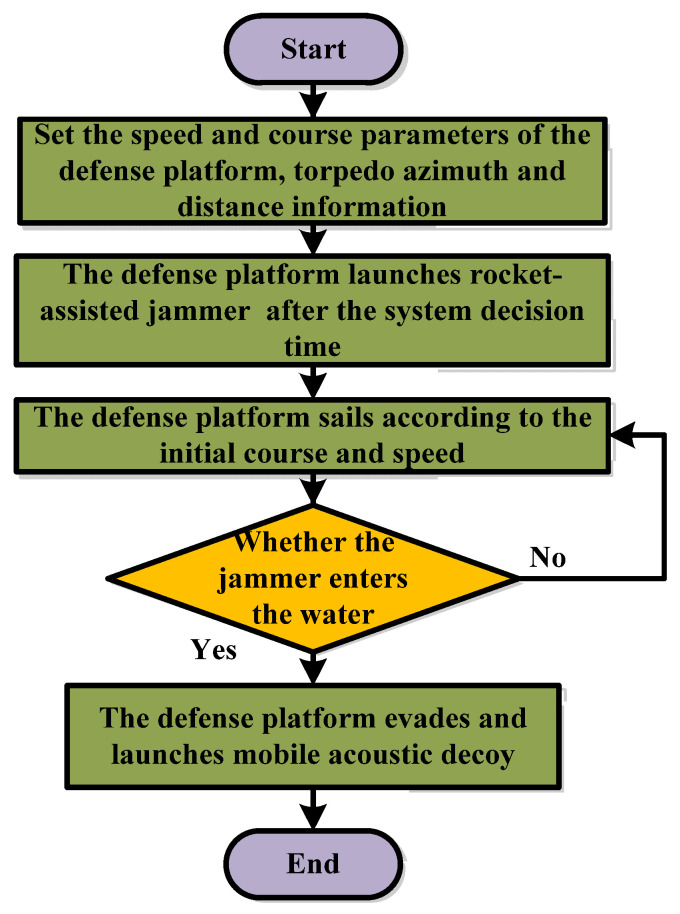
The defense platform module process.

**Figure 9 sensors-22-09701-f009:**
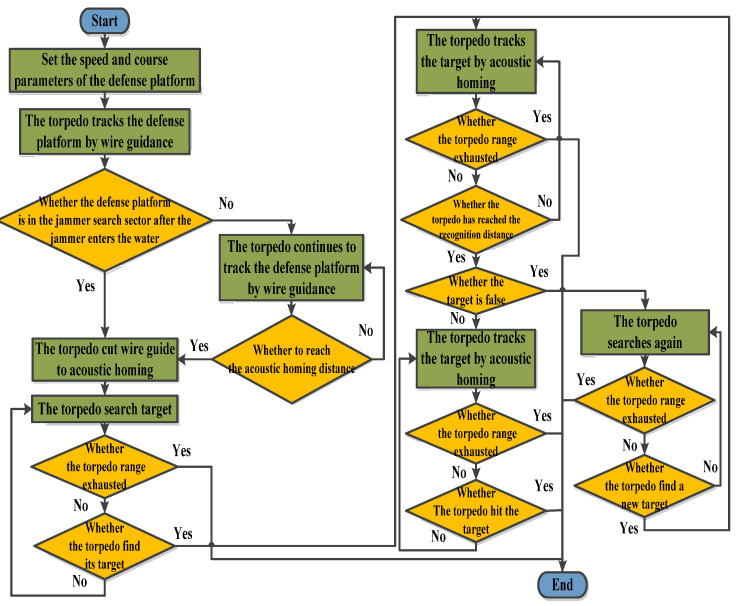
The attack platform module process.

**Figure 10 sensors-22-09701-f010:**
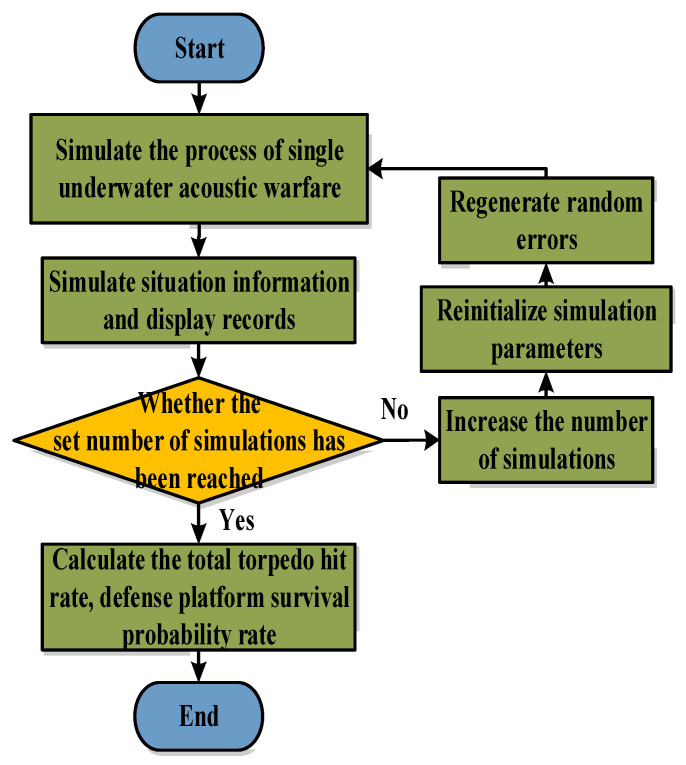
The efficiency evaluation module process.

**Figure 11 sensors-22-09701-f011:**
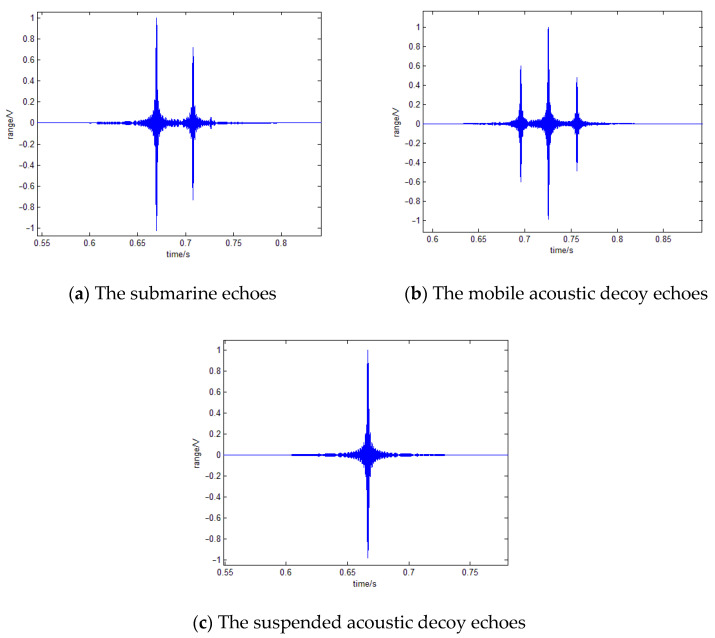
The echoes of each target at 500 m.

**Figure 12 sensors-22-09701-f012:**
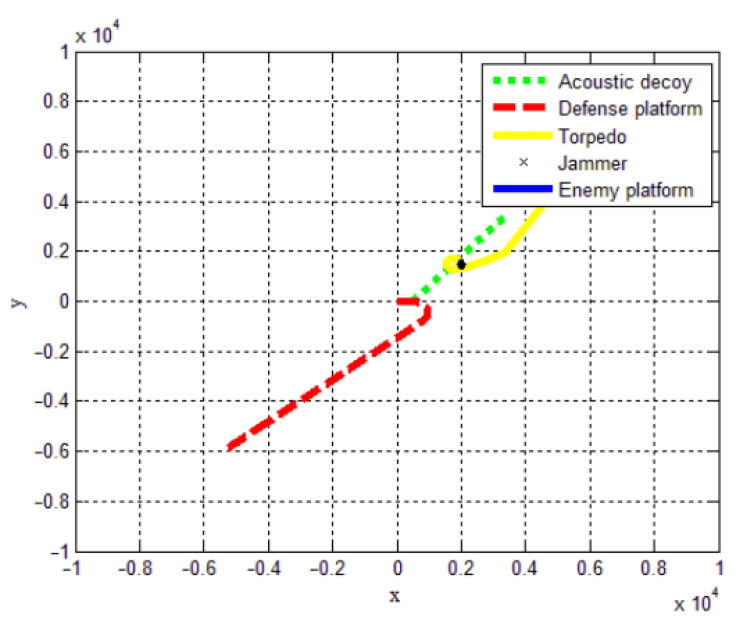
The simulation of the optimal warfare strategy.

**Figure 13 sensors-22-09701-f013:**
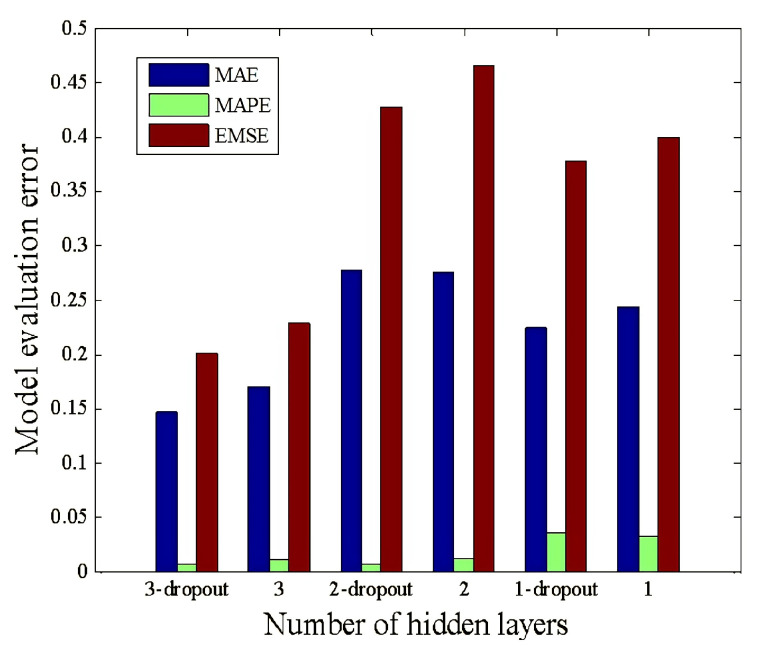
The comparison of test errors between the typical GA-BP network and the optimized GA-BP network.

**Figure 14 sensors-22-09701-f014:**
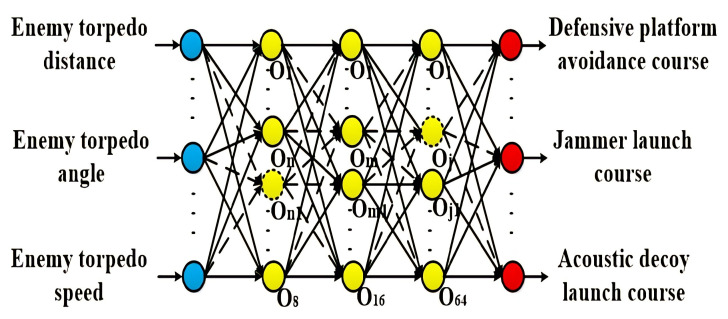
The topology of the optimized GA-BP neural network.

**Figure 15 sensors-22-09701-f015:**
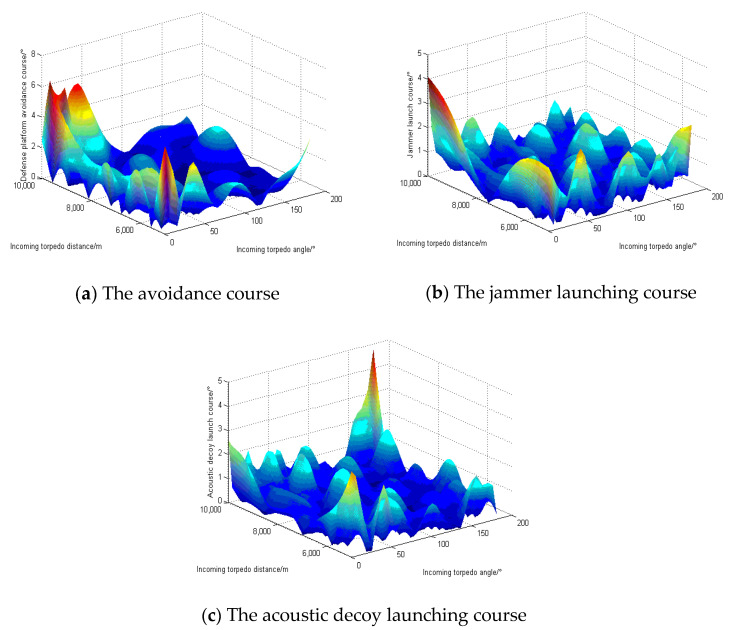
The difference between the typical GA-BP neural network’s test output and the actual output.

**Figure 16 sensors-22-09701-f016:**
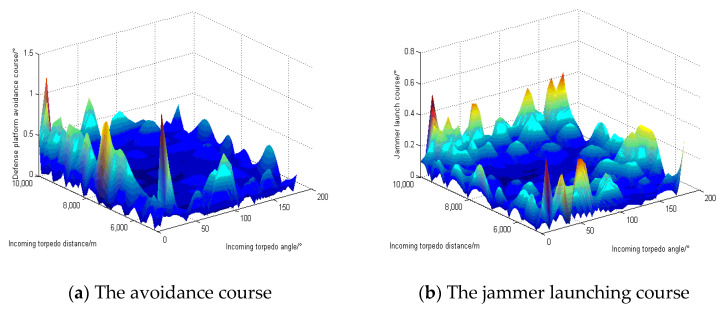
The difference between the optimized GA-BP neural network’s test output and the actual output.

**Figure 17 sensors-22-09701-f017:**
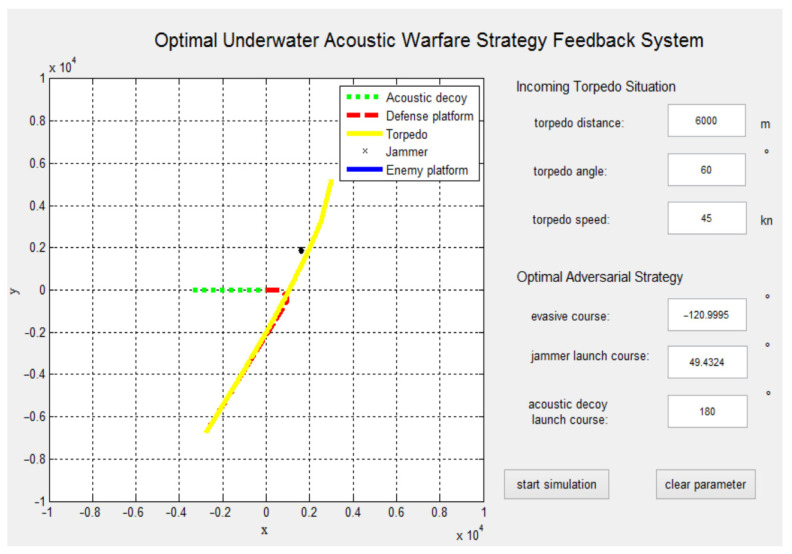
The optimal underwater acoustic warfare strategy feedback system interface.

**Table 1 sensors-22-09701-t001:** The sample set.

Sample Number	Input Samples	Output Samples
1	[*X_1_, Y_1_, Z_1_*]	*B_1_*
2	[*X_2_, Y_2_, Z_2_*]	*B_2_*
···	···	*···*
37,000	[*X_100_, Y_37_, Z_10_*]	*B_37,000_*

**Table 2 sensors-22-09701-t002:** The optimized neural network performance.

Network/ Index	MAE	MAPE	RMSE	Survival Probability
Three-layer GA-BP network after optimization	0.147	0.0061	0.2016	94.34%
Typical single-layer GA-BP network	0.2441	0.0331	0.3995	88.19%

## Data Availability

Not applicable.

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
