# Peer review of "Optimal Underwater Acoustic Warfare Strategy Based on a Three-Layer GA-BP Neural Network"

_sensors, 2022, doi:10.3390/s22249701_

Round 1

Reviewer 1 Report

I found this paper very interesting and of relevance. Although I believe any amount of improvement when it come to defense is appreciable, but still a 6.15 % seems not that  significant and so I am not quite sure how adding a three-layer GA-BP neural network adds a lot of value. 

Apart from few editorial corrections, I believe the paper can be published.

Author Response

Dear Prof:

Thank you for your very insightful comment. This letter records our responses to your comments on our manuscript entitled "Optimal hydroacoustic countermeasure strategy based on a three-layer GA-BP neural network" (ID: sensors-2012895). We have revised the title to "Optimal underwater acoustic warfare strategy based on a three-layer GA-BP neural network".

We tried our best to improve the manuscript and made some changes. These changes will not influence the content and framework of the paper. We appreciate your warm work earnestly and hope the correction will be approved.

Point 1: I found this paper very interesting and of relevance. Although I believe any amount of improvement when it come to defense is appreciable, but still a 6.15 % seems not that significant and so I am not quite sure how adding a three-layer GA-BP neural network adds a lot of value.

Apart from few editorial corrections, I believe the paper can be published.

Response 1: Thank you for your encouraging comment! We have added a description of the importance of the increased probability of survival of the defense platform in lines 20-22 and 439-445. We have explained that the method virtually eliminates the effect of human factors on defense effectiveness due to the speed and accuracy of strategy decisions.

Thank you for reading our responses and reviewing the revised manuscript!

Best regards.

Yours sincerely,

Zirui Wang.

Name: Zirui Wang.

E-mail: [email protected].

Reviewer 2 Report

For the simulation for training part, it seems there are no consideration on the marine environmental factor . How about the different between train environment and real environment , at least some discussion should be presented here.

In figure 14, The simulation of the optimal countermeasure strategy. self-platform and enemy platform should be express with other icon to seperte them from trajectories

From the training procedure , it seems building a model from state to decision parameters. It was curious that if there are model to determine this , why it was needed of Bp-networks ? 

In the conclusion part, , in line 379-381, ... poor accuracy and robustness of making hydroacoustic countermeasure strategy decisions by database search .there seems no clearly present how to achieve the robustness.

Author Response

Dear Prof:

Thank you for your very insightful comment. This letter records our responses to your comments on our manuscript entitled "Optimal hydroacoustic countermeasure strategy based on a three-layer GA-BP neural network" (ID: sensors-2012895). We have revised the title to "Optimal underwater acoustic warfare strategy based on a three-layer GA-BP neural network".

We tried our best to improve the manuscript and made some changes. These changes will not influence the content and framework of the paper. We appreciate your warm work earnestly and hope the correction will be approved.

Point 1: For the simulation for training part, it seems there are no consideration on the marine environmental factor. How about the different between train environment and real environment, at least some discussion should be presented here.

Response 1: Thank you for your very insightful comment. We have added the consideration of target state, launch signal, and ocean environment in the training part of the simulation in lines 258-282, thus making the simulation more complete. And we have added simulations of the different target echoes: Figure 11 in lines 282-283.

Point 2: In figure 14, The simulation of the optimal countermeasure strategy. self-platform and enemy platform should be express with other icon to seperte them from trajectories.

Response 2: Thank you for your precious comment. We have modified the icon expressions in Figure 12 and Figure 17 to make them more easily distinguishable. We have used solid blue lines for attack platforms, yellow lines for torpedoes, and red dashed lines for defense platforms in lines 314 and 404.

Point 3: From the training procedure, it seems building a model from state to decision parameters. It was curious that if there are model to determine this, why it was needed of Bp-networks?

Response 3: Thank you for your very insightful comment. We have added a description of the need for a GA-BP neural network decision model in lines 293 and 299. Among them, we explain that the GA-BP neural network model built on the Monte Carlo method makes decisions much faster than the Monte Carlo model. Thus this method improves the defense effectiveness of defense platform decisions.

Point 4: In the conclusion part, in line 379-381, ... poor accuracy and robustness of making hydroacoustic countermeasure strategy decisions by database search.there seems no clearly present how to achieve the robustness.

Response 4: Thank you for your very precious comment. We apologize for the lack of precision in our expression, where we should have used "stability" instead of "robustness". We have added stability validation of underwater acoustic warfare strategy decisions to demonstrate the excellent stability of the 3-layer GA-BP neural network decision system in lines 390 and 402. If we decide on the database, we will not be able to make an accurate decision when there is input data other than the database. Therefore the underwater acoustic warfare strategy decision will have poor stability.

Thank you for reading our responses and reviewing the revised manuscript!

Best regards.

Yours sincerely,

Zirui Wang.

Name: Zirui Wang.

E-mail: [email protected].
